# Geometry-Guided Generative Representation for Functional Brain Graphs

**Subati Abulikemu** [1]  **Tiago Azevedo** [1]  **Michail Mamalakis** [† 1]  **John Suckling** [† 1]

## Abstract

In network neuroscience, functional brain systems are often characterized using separate yet related graph-theoretic or spectral descriptors, overlooking how these properties covary and partially overlap across individuals and conditions. We anticipate that dense, weighted functional connectivity graphs lie on a low-dimensional latent geometry along which both topological and spectral structures vary smoothly at the population level. Although graph-based deep learning offers a powerful framework for modeling these brain connectomes, supervised approaches are constrained by the limited availability of labeled data. Existing unsupervised graph representation methods also typically focus on node-level embeddings, which are limited in capturing compact graph-level representations that preserve information from dense functional connectomes. To address these gaps, we learn compact brain graph representations using a graph transformer autoencoder, where domain-specific, aligned functional gradient geometry provides an inductive bias to guide learning. Despite being trained in a fully unsupervised manner, our approach meaningfully separates cognitive states and enables decoding of visual stimuli, with performance further improved by incorporating neural dynamics. In parallel, to enable generation of synthetic brain graphs, we fit a diffusion model to the learned latent representation and decode samples back to dense connectomes.

## 1. Introduction

Large-scale functional brain systems are modeled as weighted graphs with edges defined as statistical dependencies between distributed neural signals, i.e., functional connectivity (FC) (Biswal et al., 1995; Hallquist & Hillary, 2018). Functional connectomes display systematic variations across individuals and conditions, enabling understanding of the neural bases of cognition and its disruption. In network neuroscience, classical graph-theoretic methods have effectively revealed the small-world, modular, and hub-dominance architecture of FC graphs (Achard et al., 2006; Meunier et al., 2010). Complementarily, graph spectral decomposition has identified low-frequency, computationally meaningful eigenmodes anchoring functional organization, i.e., functional gradients (Margulies et al., 2016). Crucially, topological and spectral statistics are deterministic functions of the same FC matrix (Chung, 1997; Newman, 2006), so representation learning of dense FC graphs should preserve both organizational lenses along the learned latent manifold.

Current graph learning in connectomes, nevertheless, is largely discriminative, optimized for classification, and does not learn a latent space that represents the intrinsic geometry of brain organization at the population level (Li et al., 2021; Mohammadi & Karwowski, 2024; Thapaliya et al., 2025). Supervised signals of fMRI data are further constrained by the shortage and noise in labels (Zhang et al., 2023). Most existing unsupervised graph autoencoding computes node-level embeddings, challenged by learning a unique graph-level representation that is both compact and decodable to a full connectome (Kipf & Welling, 2016; Krzakala et al., 2025). Moreover, although FC is naturally dense and weighted, most strategies apply fixed-density thresholding, which can obscure global weight geometry and produce unstable, threshold-sensitive group comparisons (Garrison et al., 2015; van Wijk et al., 2010). Hence, we aim for an unsupervised, graph-level representation learned directly from dense FC.

Directly embedding dense, weighted FC graphs, however, comes with distinct challenges compared to standard sparse graph learners. Aggregation on message passing graph neural networks (MPNNs) relies on mixing local neighborhoods. When the graph is sufficiently large and dense, and with repeated propagation, this suffers from over-smoothing of nodal information and representational collapse (Li et al., 2018; Oono & Suzuki, 2020). Unlike MPNNs, graph transformers enable global interactions of all node-pairs, incorporating long-range dependencies essential in brain systems (Dwivedi & Bresson, 2020; Ying et al., 2021). However,

---

[†] Equal senior authorship. [1] University of Cambridge, Cambridge, United Kingdom. Correspondence to: Subati Abulikemu <ss2905@cam.ac.uk>.

*Proceedings of the 43$^{rd}$ International Conference on Machine Learning*, Seoul, South Korea. PMLR 306, 2026. Copyright 2026 by the author(s).

plain self-attention on node-pair similarities is agnostic to graph structure, and struggles to encode topological distinctions across nodes without explicit inductive bias (Dwivedi & Bresson, 2020; Ma et al., 2021; 2023; Rampášek et al., 2022). Sparse attention, on the other hand, restricts the attention to a subset of node pairs based on predefined or learned pruning schemes, which reintroduces the potential risk of discarding functionally meaningful connections (Dimitrov, 2025; Rampášek et al., 2022). For FC autoencoding, we thus retain dense interactions with an edge-conditioned encoder, and inject structure with aligned functional gradients as low-frequency geometric coordinates (Ma et al., 2023; Margulies et al., 2016). A memory-based cross-attention is then used for decoding dense FC by routing graph latent through learned node memories. Unlike existing geometry-guided graph learning, which uses graph-structural encodings (Rampášek et al., 2022), external spatial coordinates (Liu et al., 2022; Fang et al., 2022), or non-Euclidean representation spaces (Chami et al., 2019), the geometry here is connectome-specific and functionally derived from each subject's FC, then aligned across subjects to support cross-graph comparison.

Following the graph-level latent representation, we pursue three extensions. First, we augment the connectome graph autoencoder framework with neural dynamics to enable a joint spatial-temporal representation. Neural population code resides within high ambient dimensions characterized by redundancy, from which low-dimensional latent signals encoding the most essential dynamics can be derived (Cunningham & Yu, 2014; Duncker & Sahani, 2021; Hennig et al., 2018). Here, we directly condition the temporal trajectories learned via recurrent neural networks (RNNs) on the latent spatial geometry, and assess its impact on decoding cognitive states. Second, we conduct empirical investigations linking task-evoked reconfigurations of functional geometry in the latent space with cognition. Third, we enable FC generation. We model the distribution of the learned embeddings with latent diffusion rather than operating on raw graphs (Rombach et al., 2022; Zhou et al., 2024). This mitigates challenges of non-Gaussian edge distributions and edgewise generation burdens, while allowing traversal within the interpretable latent geometry before decoding to dense FC graphs.

Together, our main contributions are

- We introduce a geometry-guided transformer autoencoder that learns graph-level latent representations of *dense, weighted* FC without sparsification, with aligned functional gradients as inductive bias.

- We show that the resulting unsupervised latent representation is functionally meaningful and supports cognitive decoding, with further improvements from a spatial-temporal fusion.

- For generation, we fit a diffusion prior over the latent space to generate synthetic FC graphs that is validated in both latent and graph space.

- We utilize the latent geometry to explore task-induced reconfiguration and relate both global and directional variations to cognition. An overview of the framework is shown in Fig. 1.

## 2. Methods

### 2.1. Problem Set-up

Each individual brain (subject $\times$ session) is defined as a dense graph $\mathcal{G} = (C, X)$ with weighted connectivity $C \in \mathbb{R}^{N \times N}$ and node features $X \in \mathbb{R}^{N \times F}$. Nodes correspond to a fixed brain parcellation, therefore have consistent ordering across graphs. To construct FC matrices, fMRI time series are first per-region normalized to zero mean and unit variance, before taking pairwise Pearson correlation to obtain symmetric matrix $C$ without thresholding (Fig. 1A).

Our objective is to learn a compact graph-level latent representation $z_g \in \mathbb{R}^{d_g}$ in a fully unsupervised manner through a transformer-based autoencoder. Formally, we learn an encoder–decoder pair

$$z_g = f_\phi(C, X), \qquad \hat{C} = g_\theta(z_g), \qquad (1)$$

to minimize deterministic connectome loss with mean squared error (MSE)

$$\mathcal{L}(\phi, \theta) = \mathcal{L}_{\text{MSE}}(C, \hat{C}). \qquad (2)$$

**Functional gradients as node features** The node features $X$ of focus are diffusion-map functional gradients extracted first individually for each FC matrix then aligned across graphs (Fig. 1A) (Coifman & Lafon, 2006; Margulies et al., 2016; Vos de Wael et al., 2020). Given an FC matrix $C$, let $c_i = C_{i,:}$ be the connectivity profile of node $i$. Following common practice in functional gradient extraction, to construct the diffusion operator, for each node $i$ we retain the top 20% strongest edges in $c_i$, producing a cleaner and more stable functional geometry. Of note, this step is only conducted for gradient computation, while the embedding objective remains as dense FC. Following this, an affinity matrix $A \in \mathbb{R}^{N \times N}$ is first constructed using the normalized angle kernel

$$A_{ij} = 1 - \frac{\arccos\left(\frac{\langle c_i, c_j \rangle}{\|c_i\|\|c_j\|}\right)}{\pi} \in [0, 1], \qquad (3)$$

which represents the pairwise similarity of connectivity profiles across nodes. The degree matrix $D_{ii} = \sum_j A_{ij}$ is extracted, and the $\alpha$-normalized kernel

$$K = D^{-\alpha} A D^{-\alpha}, \qquad \alpha = 0.5, \qquad (4)$$

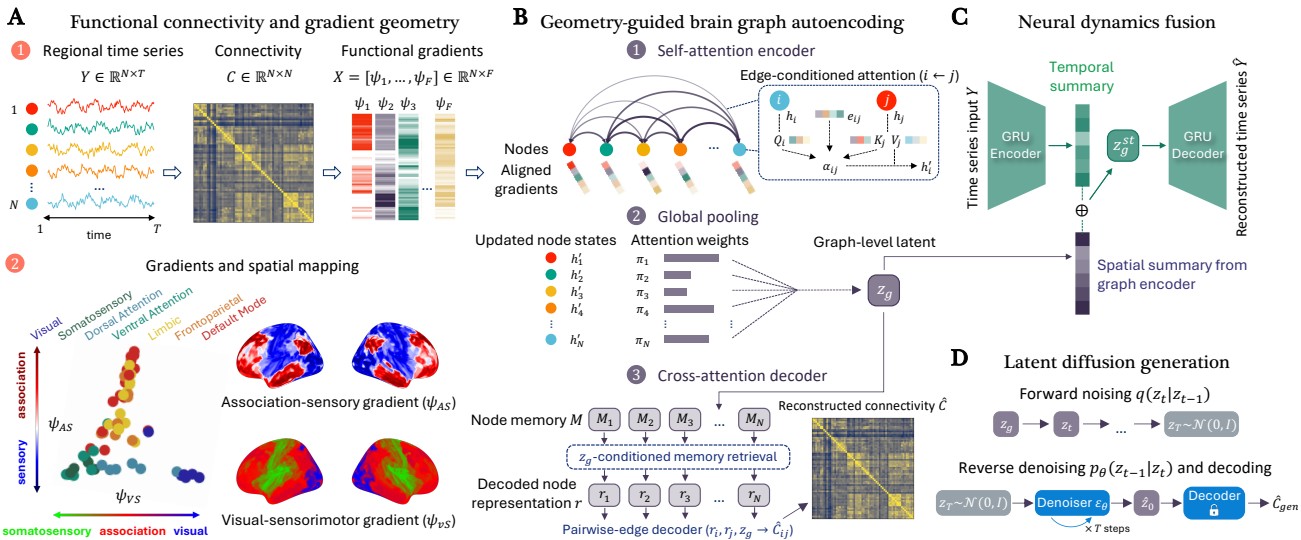

*Figure 1.* **Overview of the geometry-guided generative framework for functional brain graphs.** **(A)** Regional fMRI time series $Y$ are converted into dense FC matrix $C$, from which diffusion-map embeddings are aligned to produce functional gradients $X = [\psi_1, \ldots, \psi_F]$ summarizing macroscale brain organization. The first two gradients $\psi_{AS}$ (association–sensory) and $\psi_{VS}$ (visual–sensorimotor) are shown on the cortical surface with their canonical network composition. **(B)** The graph autoencoder maps $(C, X)$ to a graph-level latent $z_g$ and reconstructs $\hat{C}$ in three stages: **(1)** an edge-conditioned self-attention encoder that updates node states $h'_i$ using FC weights to modulate attention; **(2)** global attention pooling that aggregates node states into $z_g$; **(3)** a cross-attention decoder that retrieves from a learned node memory $M$ conditioned on $z_g$ and reconstructs connectivity through a pairwise-edge decoder. **(C)** For temporal data, a GRU encoder summarizes neural dynamics and is fused with the spatial summary from the graph encoder to form a joint spatial-temporal latent $z_g^{\text{st}}$, which conditions a GRU decoder in reconstructing $\hat{Y}$. **(D)** A denoising diffusion model is trained on $z_g$; reverse denoising from Gaussian noise gives $\hat{z}_0$, which is decoded by the frozen graph decoder to produce synthetic connectomes $\hat{C}_{\text{gen}}$.

approximating Fokker–Planck diffusion is constructed. Given the Markov diffusion operator

$$P = \left(\sum_j K_{ij}\right)^{-1} K, \qquad (5)$$

its eigenvectors $\{\phi_k\}$ define the functional gradients. Each gradient is rescaled by a multiscale weighting that considers contributions across all diffusion times ($t = 1, 2, \ldots$). Here, we obtain the multiplier $\sum_{t=1}^{\infty} \lambda_k^t = \lambda_k/(1 - \lambda_k)$ and final gradient $\psi_k = \frac{\lambda_k}{1-\lambda_k}\phi_k$ (Richards et al., 2009; Vos de Wael et al., 2020), resulting

$$X = [\psi_1, \ldots, \psi_F] \in \mathbb{R}^{N \times F}. \qquad (6)$$

After obtaining individual embeddings, they are aligned to a reference template via orthogonal Procrustes. This preserves within-subject geometry while ensuring directional correspondence across graphs. Crucially, the reference template is computed strictly from the resting-state *training set* by extracting gradients from the mean FC matrix, preventing data leakage. The same template is then used to align gradients for all graphs (resting-state and task, including training/validation/test sets).

The network organizational differentiation along each aligned gradient axis can be summarized by its diffusion range

$$\text{range}(\psi_k) = \max \psi_k - \min \psi_k. \qquad (7)$$

Empirically, $\text{range}(\psi_k)$ is highly correlated with $\lambda_k$ across graphs (see Appendix). After alignment, however, eigenvalues are not uniquely associated with a specific aligned gradient axis due to subspace rotations. Therefore, we use $\text{range}(\psi_k)$ as a direct, sign-invariant measure of differentiation along the aligned coordinate.

## 2.2. Edge-Conditioned Self-Attention Encoder

In graph encoding, we map $(C, X) \mapsto z_g$ with $L_e$ edge-conditioned transformer layers similar to Ma et al. (2023) (Fig. 1B, step 1). Let $d_h$ represent node hidden dimension, $d_e$ edge hidden dimension, and $H$ the number of attention heads with $d_k = d_h/H$. Both node $h^{(\ell)} \in \mathbb{R}^{N \times d_h}$ and edge tokens $e^{(\ell)} \in \mathbb{R}^{N \times N \times d_e}$ are held throughout the encoding process.

Node and edge representations are initialized by linear projections

$$h^{(0)} = XW_{\text{init}}, \qquad e_{ij}^{(0)} = C_{ij}E_{\text{init}}, \qquad (8)$$

where $E_{\text{init}}$ projects scalar FC weights to $d_e$-dimensional edge features.

**Edge-conditioned attention**  At encoder layer $\ell$, queries, keys, and values are computed from node tokens

$$Q_i = h_i^{(\ell)} W_Q, \qquad K_i = h_i^{(\ell)} W_K, \qquad V_i = h_i^{(\ell)} W_V, \tag{9}$$

followed by constructing edge-conditioned logit vectors

$$\hat{e}_{ij} = \text{GELU}\Big(\rho\big((Q_i + K_j) \odot (e_{ij}^{(\ell)} E_w)\big) + (e_{ij}^{(\ell)} E_b)\Big), \tag{10}$$

$$\alpha_{ij} = \text{softmax}_j \left( \frac{\hat{e}_{ij} \cdot w_A}{\sqrt{d_k}} \right), \tag{11}$$

where $\odot$ is Hadamard product and $\rho(x) = \text{sign}(x)\sqrt{|x| + \epsilon}$ stabilizes the edge modulation. $E_w$ and $E_b$ are learned linear maps projecting the edge token $e_{ij}^{(\ell)}$ to $d_k$ dimensions, $w_A$ is a learned per-head projection that reduces $\hat{e}_{ij} \in \mathbb{R}^{d_k}$ to a scalar attention logit. We then aggregate edge-conditioned values as $m_i = \sum_{j=1}^{N} \alpha_{ij} \Big( V_j + \hat{e}_{ij} E_v \Big)$, where $E_v$ projects $\hat{e}_{ij}$ to the value dimension, and $m_i$ is the message used to update the node state $h_i$. Multi-head messages are updated through residual connections and feedforward networks for both node and edge representations.

**Graph pooling**  After $L_e$ layers we obtain node states $h^{(L_e)} \in \mathbb{R}^{N \times d_h}$ and pool them into a compact graph summary through global attention (Fig. 1B, step 2)

$$\pi = \text{softmax}\big(\tanh(h^{(L_e)} W_{\text{pool}}) \cdot c_{\text{pool}}\big), \tag{12}$$

$$s = \sum_{i=1}^{N} \pi_i h_i^{(L_e)}, \qquad z_g = W_z s + b_z. \tag{13}$$

This encodes a graph-level embedding $z_g$ capturing subject-specific functional organization.

## 2.3. Cross-Attention Graph Decoder

For decoding, we reconstruct $\hat{C}$ from the compressed $z_g$ using a *memory mechanism* (Fig. 1B, step 3). We define a learnable memory $M \in \mathbb{R}^{N \times d_m}$, where $M_i$ is a persistent embedding for node $i$. The memory acts as a shared prior over regional characteristics, and this allows $z_g$ to encode graph-specific modulation. Cross-attention uses $z_g$ as a routing signal that selectively retrieves and combines these priors to instantiate a graph-specific realization.

**Cross-attention over node memory**  Keys and values are constructed from the memory as $K = MW_K$ and $V = MW_V$, and we initialize node state as $h^{(0)} = MW_{\text{init}}$. At decoder layer $\ell$, each node forms a query from both the graph latent and its current state

$$Q_i = W_{q,z} z_g + W_{q,h} h_i^{(\ell)}, \tag{14}$$

and attends over the memory

$$\alpha_{ij} = \text{softmax}_j \left( \frac{Q_i \cdot K_j}{\sqrt{d_k}} \right), \qquad m_i = \sum_{j=1}^{N} \alpha_{ij} V_j, \tag{15}$$

followed by residual connections and feedforward network to update $h^{(\ell)}$. After $L_d$ layers, we obtain node states $h^{(L_d)}$ and form node embedding $r_i = \phi_r(h_i^{(L_d)})$.

We reconstruct edges conditioned on $z_g$

$$\tilde{C}_{ij} = \phi_E([r_i;\, r_j;\, r_i \odot r_j;\, |r_i - r_j|;\, z_g]), \tag{16}$$

and enforce symmetry with

$$\hat{C} = \tfrac{1}{2}\left(\tilde{C} + \tilde{C}^{\top}\right). \tag{17}$$

## 2.4. Latent Diffusion on $z_g$

After the autoencoder is trained, we train a denoising diffusion probabilistic model (Ho et al., 2020) on $z_g$ (Fig. 1D) with a linear noise schedule $\{\beta_t\}_{t=1}^{T}$: $q(z_t | z_{t-1}) = \mathcal{N}(\sqrt{1 - \beta_t}\, z_{t-1}, \beta_t I)$, with the denoising network $\epsilon_\theta$ trained on $\mathbb{E}_{t, z_0, \epsilon} \|\epsilon - \epsilon_\theta(z_t, t)\|^2$. Sampling runs the reverse process to obtain $z_0$; the decoder was *frozen* to map $z_0 \mapsto \hat{C}$.

## 2.5. Neural Dynamics Extension

For a subject with static FC graph $(C, X)$ and neural time series matrix $Y_{1:T} \in \mathbb{R}^{N \times T}$, we extend the encoder to a dual pathway that combines spatial structure and neural activity into a joint spatial-temporal representation $z_g^{\text{st}}$, then decode $Y_{1:T}$ with a Recurrent Neural Network (RNN) conditioned on this representation (Fig. 1C). The aim is to learn a low-dimensional dynamical system of neural activities that is directly modulated by $z_g^{\text{st}}$.

**Dynamics extractor and fusion**  We take $Y_{1:T}$ as a sequence of whole-brain activation vectors, and process them with an encoder RNN [Gated Recurrent Unit (GRU)]. At each time step, the input to GRU is the $N$-dimensional vector $y_t = Y_{:,t} \in \mathbb{R}^N$. Encoder GRU processes this sequence and maintains a hidden state $h_t$ as a global temporal summary up to time step $t$. These hidden states are aggregated with temporal mean, $h_{\text{time}} = \frac{1}{T} \sum_{t=1}^{T} h_t$, producing a single dynamics summary. This summary is transformed to the same graph-level embedding space as the spatial encoder through $s_{\text{time}} = \phi_{\text{time}}(h_{\text{time}})$. In parallel, the graph encoder gives a spatial summary $s_{\text{space}}$ (Section 2.2). We concatenate and fuse these representations, $\phi_{\text{fuse}}([s_{\text{space}} \,\|\, s_{\text{time}}])$, to derive the joint latent $z_g^{\text{st}}$.

**Temporal decoder**  With the temporal decoding, we aim to unroll a low, $r$-dimensional latent trajectory from $z_g^{\text{st}}$

before projecting onto the node space. We condition the decoder GRU using an initial condition $h_0$ and a constant context $c$, both derived from $z_g^{\text{st}}$

$$h_0 = W_{\text{ic}}\, z_g^{\text{st}} + b_{\text{ic}}, \qquad c = W_{\text{ctx}}\, z_g^{\text{st}} + b_{\text{ctx}}. \tag{18}$$

Latent neural dynamics $\{h_t\}_{t=1}^T$, $h_t \in \mathbb{R}^r$, is evolved with

$$h_t = \text{GRUCell}(c, h_{t-1}). \tag{19}$$

A linear readout reconstructs the neural signal at each time step

$$\hat{Y}_{:,t} = L\, h_t + b \quad \Longrightarrow \quad \hat{Y} \in \mathbb{R}^{N \times T}. \tag{20}$$

## 3. Experiments

### 3.1. Experimental Setup and Evaluation Protocol

For our main experiments, we used fMRI data from the Human Connectome Project (HCP; $N = 1067$; Van Essen et al., 2013), including resting-state and task fMRI across seven cognitive tasks acquired in separate sessions. Subjects were split by IDs into 70/10/20 training/validation/test sets. We followed the leakage-free alignment protocol in Section 2.1: the functional-gradient template was derived from the resting-state training set and used to align gradients for all graphs. For clinical evaluation, we used the Bipolar and Schizophrenia Network for Intermediate Phenotypes (BSNIP; $n = 984$; Tamminga et al., 2014) with 5-fold cross-validation stratified by diagnostic label (10% held out for validation within each fold).

We use the first $F_{\text{rest}} = 10$ aligned functional gradients as node features for resting-state experiments and the first $F_{\text{task}} = 30$ aligned gradients for task-state experiments; this convention is used throughout the main results, baselines, and ablations. Main experiments used a 64-region parcellation, with 100- and 150-region parcellations in ablations (Section 3.5). We used $d_g = 16$ for resting-state models and $d_g = 32$ for task-state models for the graph latent $z_g$.

Our first set of experiments tested the efficacy of functional gradients as an inductive bias for learning dense, weighted FC graphs in both resting-state and working-memory (WM) data (Section 3.2), and evaluated whether the resulting unsupervised graph-level representation $z_g$ supports cognitive decoding, with and without the temporal extension (Section 3.2.3). For WM, each subject's time series was partitioned into eight task blocks (2 loads × 4 stimulus types), which have equal temporal length by design, enabling RNN modeling. Next, using all seven tasks, we related subject-level variations in the task-based latent neural geometry to behavioral performance and cognition (Section 3.3). For generation, we modeled the latent distribution with diffusion and assessed whether it can sample realistic synthetic brain

graphs (Section 3.4). Finally, we compared against relevant graph-representation baselines and performed ablations (Section 3.5), evaluating both reconstruction and latent separability of cognitive states and clinical condition. Across sections, we report reconstruction with edgewise MSE and generative validation with distributional alignment in latent and graph space. For decoding, the autoencoder is trained only on the reconstruction objective without label supervision; downstream classifiers are fit post-hoc on frozen $z_g$ using logistic regression, under the same subject-wise split used for unsupervised training. Code is available at github.com/SubatA20/geometry-guided-brain-graph-AE.

### 3.2. Functional Brain Graph Representation

#### 3.2.1. SPECTRAL INDUCTION IN BRAIN GRAPH LEARNING

Using resting-state fMRI from HCP, we trained the same autoencoder while varying node features, including (1) diffusion-map spectral embeddings with and without alignment (Margulies et al., 2016); (2) node-level graph-theoretic features (strength, clustering and participation coefficients, eigenvector centrality, betweenness centrality, local efficiency, and within-module $z$-score); and (3) an edge-only control with constant node features. We also compared against a graph convolutional autoencoder (GAE; graph convolutional encoder with a multi-layer perceptron [MLP] decoder) using diffusion-map embeddings as node attributes.

Overall, alignment substantially enhanced the utility of functional gradients, with the full model achieving the best reconstruction of dense FC (Fig. 2A; MSE = $0.0124 \pm 0.0006$, mean $\pm$ standard deviation [SD] over four seeds). This outperformed the unaligned gradient ablation ($0.0172 \pm 0.0005$), graph-theoretic features ($0.0153 \pm 0.0003$), edge-only encoding ($0.0161 \pm 0.0007$), and the GAE baseline ($0.0177 \pm 0.0009$). Additional baselines and ablations are reported in Section 3.5.

#### 3.2.2. STRUCTURED VARIATION IN LATENT SPACE

As a diagnostic analysis, we examined whether the learned graph-level representation $z_g$ preserves coupled spectral and graph-theoretic lenses of network organization. Because such summaries are deterministic functions of the same FC matrix (and gradient spreads are derived from diffusion-map embeddings as node features), we assess what structure is retained under a low-dimensional bottleneck. In resting state, as shown in Fig. 2B–C, subjects exhibited graded variations in both spectral (association–sensory $\psi_{AS}$, visual–sensorimotor $\psi_{VS}$ diffusion spread) and graph properties (mean strength, modularity, small-worldness) along the principal directions of $z_g$, motivating a low-rank characterization.

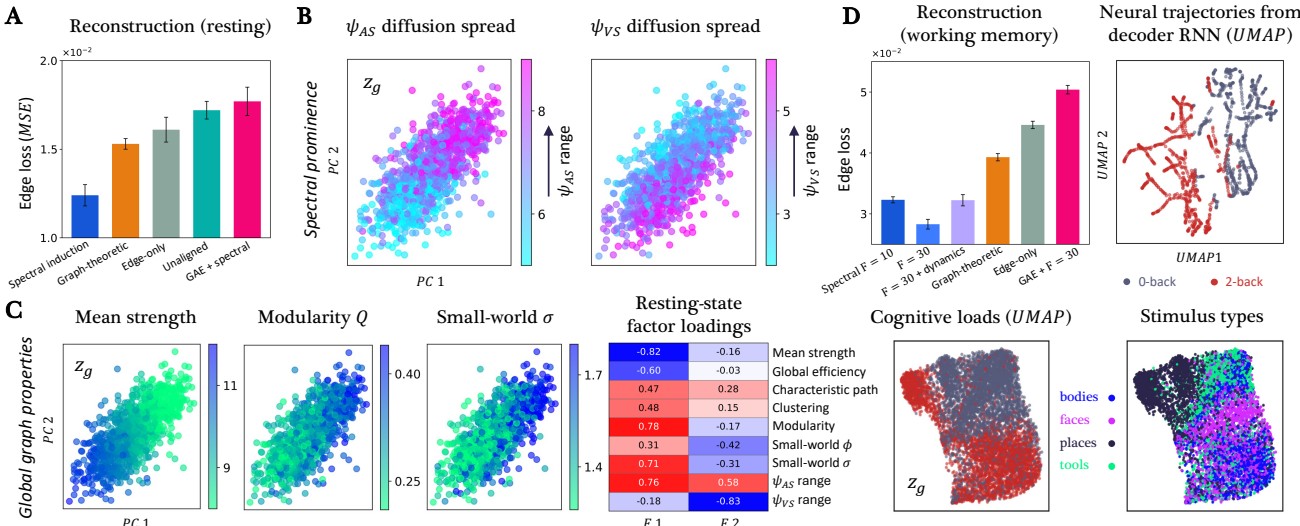

*Figure 2.* **Unsupervised learning of dense functional connectomes.** **(A)** Resting-state edge reconstruction MSE (mean ± SD over four seeds) for the graph-transformer autoencoder under different node-feature sets (aligned functional gradients, edge-only constant features, and graph-theoretic features), and a GAE baseline using aligned gradients. **(B)** Subject embeddings $z_g$ (first two PCs) vary smoothly with diffusion range of the association–sensory gradient ($\psi_{AS}$) and the visual–sensorimotor gradient ($\psi_{VS}$). **(C)** Latent embeddings $z_g$ show graded variation in mean strength, modularity $Q$, and small-world $\sigma$, with a two-factor readout summarizing loadings of global graph and gradient-range metrics. **(D)** Working memory edge reconstruction MSE (mean ± SD over four seeds) across node-feature sets, spatial-temporal model, and GAE. UMAP visualizations show decoder RNN neural trajectories (top) and fused spatial-temporal embeddings $z_g^{\mathrm{st}}$ (bottom), separating cognitive load (0-back vs. 2-back) and stimulus category (bodies/faces/places/tools).

To assess whether a small number of latent directions capture distinct patterns of metric covariation, we fitted a reduced-rank multivariate linear regression from $z_g$ to the nine normalized metrics (seven global graph-theoretic metrics, including mean strength, global efficiency, characteristic path length, mean clustering, modularity $Q$, small-world $\phi$, small-world $\sigma$, and two interpretable gradient-spread measures for $\psi_{AS}$ and $\psi_{VS}$). With $Z$ representing subject embeddings and $Y$ the metric matrix, we modeled $Y \approx ZB$ with $\mathrm{rank}(B) \leq K$, or equivalently $\hat{Y} = (ZA)C^\top$, where the columns of $A$ define $K$ directions in $z_g$ and the columns of $C$ define the corresponding metric profiles. We selected $K$ via 5-fold cross-validation on the training set as the smallest rank achieving mean held-out multivariate $R^2 \geq 0.50$, which gave $K = 2$. On the test set, this two-factor readout explained $R^2 = 0.52$ of the variance across the nine metrics ($\Delta R^2 = \{0.38, 0.14\}$). The first factor associated larger $\psi_{AS}$ spread with increased segregation (higher small-worldness and modularity) and decreased integration (lower mean strength and global efficiency), whereas the second factor contrasted $\psi_{AS}$ and $\psi_{VS}$ and captured a weaker residual small-world pattern associated with $\psi_{VS}$ spread.

### 3.2.3. COGNITIVE STATES AND TEMPORAL EXTENSION

On the more heterogeneous task-state FC graphs from the working-memory (WM) task, the advantages of functional gradient induction and our architecture were amplified. Using $F = 10$ and $F = 30$ gradients achieved reconstruction

MSE of $0.0323 \pm 0.0005$ and $0.0283 \pm 0.0008$, respectively, outperforming graph-theoretic features ($0.0393 \pm 0.0006$), edge-only encoding ($0.0446 \pm 0.0006$), unaligned gradients at $F = 30$ ($0.0462 \pm 0.0012$), and the GAE baseline ($0.0504 \pm 0.0004$; Fig. 2D).

Despite being trained without labels, the functional-gradient induction model ($F = 30$) learned embeddings $z_g$ that captured meaningful cognitive structure. A logistic regression trained on $z_g$ and evaluated on the held-out test set classified 0-back vs. 2-back load at 78.6% (AUC 0.862), and decoded visual stimulus category (bodies/faces/places/tools) at 60.4% (0-back) and 64.2% (2-back). Incorporating neural dynamics via the dual spatial-temporal autoencoding pathways, the fused representation $z_g^{\mathrm{st}}$ was more informative; load decoding improved to 89.7% (AUC 0.954), and stimulus decoding to 84.5% (AUC 0.961) and 75.2% (AUC 0.919) under 0-back and 2-back, respectively (Fig. 2D).

### 3.3. Neural Geometry Reconfiguration and Cognitive Function

#### 3.3.1. TASK-STATE DISPERSION AND COGNITIVE PERFORMANCE

When embedding FC graphs from the seven cognitive tasks, a logistic regression on the task-based $z_g$ achieved 84.7% accuracy (AUC 0.973; Table 1). The task-fMRI latent geometry was then linked to behavior, to explore how subject-level $z_g$ variations associate with task performances and general

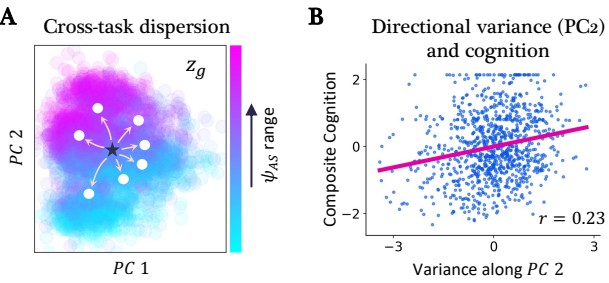

*Figure 3.* **Neural geometry reconfiguration and cognition. (A)** Task-state embeddings $z_g$ are projected onto the first two PCs, color represents $\psi_{AS}$ range. For each subject, global dispersion is defined as the mean Euclidean distance of task embeddings to the subject centroid in the original latent space (white dots show example task embeddings; star indicates the centroid). **(B)** Subject-level variance along PC2 ($\psi_{AS}$ gradient-related axis) is positively associated with composite cognition ($r = 0.23$).

cognition. For each subject and task, the task-specific dispersion was computed as the Euclidean distance of the task embedding to the subject's mean embedding (centroid) in $z_g$ (Fig. 3A), with which we regressed the task accuracy and reaction time controlling for age and gender. Across working memory, relational, emotion, and language tasks, greater dispersion was associated with lower accuracy and slower reaction times (standardized $|\beta_{\text{disp}}| = 0.08$–$0.15$, FDR-adjusted $p < 0.05$). Globally, greater mean dispersion across task embeddings was also associated with lower composite cognitive score ($\beta_{\text{disp}} = -0.16$, $p < 0.001$) measured using the NIH toolbox (nihtoolbox.org). Overall, this suggests that larger multitask reconfiguration in the latent space relates to poorer cognitive functions, which is in line with previous work showing individuals with higher intelligence display less neural adaptation for specific tasks (Dunst et al., 2014; Thiele et al., 2022).

### 3.3.2. GRADIENT-RELATED RECONFIGURATIONS BENEFIT COGNITION

To understand which interpretable latent directions of task-based $z_g$ influence cognition, we further decomposed $z_g$ into principal components and related them to both network properties and cognition. PC1 was dominated by the global strength ($r = 0.95$) and efficiency ($r = 0.50$), whereas PC2 tracked predominantly the spread of association-sensory gradient ($r = 0.70$, Fig. 3A), and to a weaker extent, visual–sensorimotor gradient ($r = 0.30$) and modularity ($r = 0.20$). At the subject level, greater cross-task variance along PC2, which reflects fluctuations in the functional differentiation between higher- and lower-order systems, i.e., $\psi_{AS}$, across tasks, was positively associated with composite cognition (standardized $\beta = 0.23$, $p < 0.001$; Fig. 3B), controlling for age and gender. While the variance along the strength dominant axis (PC1) showed small negative associa-

tion ($\beta = -0.07$, $p = 0.03$). Hence, not only the overall but also directional reconfiguration of the latent representation is behaviorally relevant; diffuse dispersion across tasks may be detrimental, while modulation along a gradient-related axis seems cognitively beneficial.

### 3.4. Latent Diffusion and Brain Graph Generation

We modeled the distribution of graph-level embeddings by fitting a diffusion prior $p(z_g)$ on the encoder latent space learned from resting-state FC graphs (Fig. 4A). To evaluate whether diffusion samples robustly captured the empirical latent distribution, we compared generated and held-out test embeddings in normalized latent space (using training set normalization parameters) using two complementary metrics. First, we computed maximum mean discrepancy (MMD), which is a distance measure that approaches zero when two distributions match. Second, we applied a 1-nearest-neighbor (1-NN) two-sample test, which measures how well points can be classified as test vs. generated based on their nearest neighbor's label (chance $= 0.5$ for size-matched sets). Across 50 size-matched generated subsamples, we obtained $\text{MMD}^2 = 0.00136 \pm 0.00107$ and 1-NN accuracy $= 0.537 \pm 0.026$, indicating that generated latents closely approximated the test latent distribution.

With the dense FC graphs decoded from diffusion samples (Fig. 4B), we assessed generation quality in both matrix and graph space (Fig. 4C). In matrix space, generated graphs matched the test distribution of off-diagonal connectivity weights (Kolmogorov–Smirnov statistic KS $= 0.033$, Wasserstein distance $= 0.029$) without sparsification. In spectral domain, the distribution of the top-10 leading eigenvalues also showed good alignment with test graphs (mean KS $= 0.093$, max KS $= 0.129$), showing the preservation of dominant global spectral modes. Topologically, distributions of mean clustering coefficient, modularity, and small-world $\sigma$ were also aligned between test and generated sets (KS $= 0.084, 0.074, 0.061$, respectively).

### 3.5. Graph Representation Baselines and Ablations

**Baselines** We compared our functional connectome embedding method with several unsupervised graph representation baselines (Table 1), evaluating both dense connectome reconstruction and downstream classification using logistic regression on the learned $z_g$. Classification tasks included task-state prediction (7-way) in HCP task-fMRI and healthy control vs. schizophrenia in the BSNIP dataset. For fairness, aligned functional gradients were used as node features for all graph-based methods and the dimensionality of $z_g$ was matched across models.

For both reconstruction and latent separability, we adapted existing graph autoencoding baselines to derive a deterministic whole-brain summary $z_g$. These included (1)

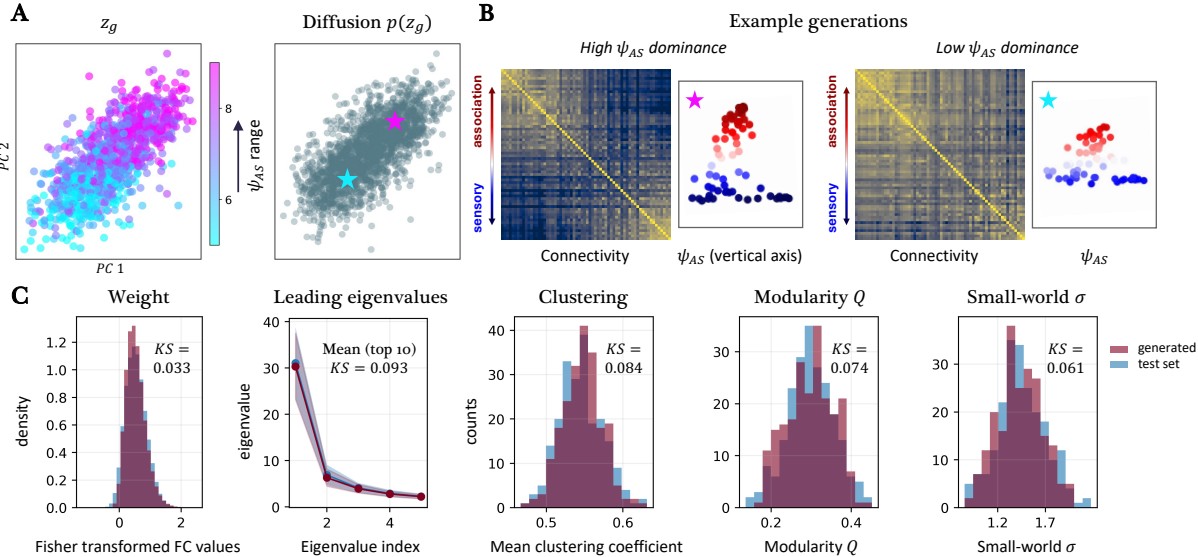

*Figure 4.* **Latent diffusion and functional connectome generation.** (**A**) Resting-state latent space $z_g$ colored by association–sensory gradient ($\psi_{AS}$) range, with samples from the diffusion-learned latent distribution $p(z_g)$. (**B**) Representative generated connectomes from high- and low-$\psi_{AS}$ regions of the learned distribution, corresponding to the starred samples in panel A. Connectivity matrices are ordered along the association–sensory axis, with diffusion-map embeddings of the generated matrices shown to the right. (**C**) Distribution alignment of connectivity weights, leading eigenvalues, and graph statistics (mean clustering, modularity $Q$, small-world $\sigma$) between test and generated sets.

a plain *GAE* with a graph-convolutional encoder and an MLP decoder; (2) a *Graphite*-inspired model (Grover et al., 2019) with an iterative refinement decoder that constructs an intermediate adjacency from node-pair inner products and refines node representations via message passing; (3) the graph-level transformer autoencoder *GRALE* (Krzakala et al., 2025), based on Evoformer encoder and decoder, with pooled pairwise representations for graph embedding; and (4) the brain connectome-specific *GATE* (Liu et al., 2021), which encodes vectorized edges with an MLP and decodes $z_g$ into node-wise factor vectors, whose outer products are summed to reconstruct connectivity. To assess separability without graph decoding, we further included representation-only controls, including *UMAP* embeddings of vectorized FC and *GraphMAE* (Hou et al., 2022), a self-supervised masked graph autoencoder trained to reconstruct masked node attributes, using pooled encoder embeddings as $z_g$.

As summarized in Table 1, our architecture achieved the lowest reconstruction error among reconstruction-enabled baselines for FC graphs and the highest latent linear separability on both HCP tasks and BSNIP healthy controls vs. schizophrenia patients.

**Ablations** We conducted four ablations to test our design choices (Table 2). *No alignment* uses diffusion-map gradients extracted per subject without Procrustes alignment to the train-only resting-state template, removing cross-subject directional correspondence in the geometric coordinates. *No edge conditioning* replaces the edge-conditioned trans-

former encoder with a plain self-attention encoder that attends over node tokens only, such that connectivity weights are not used to modulate attention logits or values. *No memory* replaces the cross-attention decoder over a node memory with a memoryless decoder that maps $z_g$ directly for reconstruction through a global MLP (without node-wise memory retrieval). *Edge-only encoding* keeps the same encoder but uses constant node features (all ones), such that learning relies on edge weights alone rather than the gradient geometry. Across parcellations (64/100/150 nodes) and both rest and task, each ablation degraded reconstruction relative to the full model.

## 4. Conclusion and Limitations

This study presented a geometry-guided latent representation for dense, weighted functional brain graphs. Our approach combines an edge-conditioned graph transformer encoder with a memory-based cross-attention decoder to learn compact graph-level embeddings that reconstruct connectomes. Aligned functional gradients provide a domain-specific inductive bias that improves representation learning, where we derive a latent space with coherent, smoothly varying brain organizational structure across individuals. Despite being trained without labels, the unsupervised embeddings enabled robust decoding of cognitive states and stimulus categories, and incorporating neural dynamics into the spatial representation further strengthens this functional separability. Beyond representation, we model the embed-

*Table 1.* **Graph learning baselines.** Edge reconstruction (MSE ↓) and latent separability (↑). HCP results average over 4 seeds (random train/val/test splits; gradients templated from training-rest per seed); BSNIP uses 5-fold CV.

| Model | HCP Rest Recon MSE↓ | HCP Task Recon MSE↓ | Latent (7-way) ACC↑ | Latent (7-way) Macro AUC↑ | BSNIP Recon MSE↓ | Latent (HC vs. SCZ) ACC↑ | Latent (HC vs. SCZ) AUC↑ |
|---|---|---|---|---|---|---|---|
| GAE | $0.0177_{\pm0.0009}$ | $0.0209_{\pm0.0010}$ | $0.6669_{\pm0.0217}$ | $0.9206_{\pm0.0106}$ | $0.0206_{\pm0.0005}$ | $0.6278_{\pm0.0500}$ | $0.6579_{\pm0.0587}$ |
| GRAPHITE $_{Dec.}$ | $0.0156_{\pm0.0006}$ | $0.0158_{\pm0.0005}$ | $0.7979_{\pm0.0136}$ | $0.9614_{\pm0.0048}$ | $0.0182_{\pm0.0005}$ | $0.6036_{\pm0.0605}$ | $0.6506_{\pm0.0629}$ |
| GRALE | $0.0159_{\pm0.0013}$ | $0.0180_{\pm0.0009}$ | $0.7813_{\pm0.0162}$ | $0.9581_{\pm0.0059}$ | $0.0227_{\pm0.0018}$ | $0.5679_{\pm0.1026}$ | $0.5802_{\pm0.1070}$ |
| GATE $_{Connectome}$ | $0.0155_{\pm0.0008}$ | $0.0184_{\pm0.0014}$ | $0.7354_{\pm0.0313}$ | $0.9456_{\pm0.0112}$ | $0.0197_{\pm0.0004}$ | $0.6263_{\pm0.0371}$ | $0.6580_{\pm0.0308}$ |
| FC + UMAP | | | $0.7500_{\pm0.0190}$ | $0.9521_{\pm0.0050}$ | | $0.5909_{\pm0.0467}$ | $0.6222_{\pm0.0607}$ |
| GRAPHMAE | | | $0.7371_{\pm0.0092}$ | $0.9465_{\pm0.0057}$ | | $0.6081_{\pm0.0380}$ | $0.6445_{\pm0.0403}$ |
| **Ours** | $\mathbf{0.0124}_{\pm0.0006}$ | $\mathbf{0.0143}_{\pm0.0008}$ | $\mathbf{0.8465}_{\pm0.0131}$ | $\mathbf{0.9727}_{\pm0.0049}$ | $\mathbf{0.0156}_{\pm0.0003}$ | $\mathbf{0.6646}_{\pm0.0540}$ | $\mathbf{0.7370}_{\pm0.0483}$ |

*Table 2.* **Edge reconstruction** (MSE ↓) on HCP **Rest** and **Task** across parcellations (64/100/150 nodes).

| Model | HCP Rest 64 | HCP Rest 100 | HCP Rest 150 | HCP Task 64 | HCP Task 100 | HCP Task 150 |
|---|---|---|---|---|---|---|
| No alignment | $0.0172_{\pm0.0005}$ | $0.0192_{\pm0.0004}$ | $0.0206_{\pm0.0003}$ | $0.0222_{\pm0.0014}$ | $0.0239_{\pm0.0012}$ | $0.0244_{\pm0.0012}$ |
| No edge conditioning | $0.0210_{\pm0.0006}$ | $0.0236_{\pm0.0002}$ | $0.0243_{\pm0.0007}$ | $0.0184_{\pm0.0012}$ | $0.0212_{\pm0.0013}$ | $0.0229_{\pm0.0013}$ |
| No memory | $0.0168_{\pm0.0009}$ | $0.0189_{\pm0.0003}$ | $0.0201_{\pm0.0004}$ | $0.0203_{\pm0.0009}$ | $0.0222_{\pm0.0011}$ | $0.0235_{\pm0.0012}$ |
| Edge-only encoding | $0.0161_{\pm0.0007}$ | $0.0183_{\pm0.0003}$ | $0.0198_{\pm0.0005}$ | $0.0215_{\pm0.0014}$ | $0.0234_{\pm0.0013}$ | $0.0239_{\pm0.0014}$ |
| **Ours** | $\mathbf{0.0124}_{\pm0.0006}$ | $\mathbf{0.0158}_{\pm0.0006}$ | $\mathbf{0.0171}_{\pm0.0003}$ | $\mathbf{0.0143}_{\pm0.0008}$ | $\mathbf{0.0158}_{\pm0.0007}$ | $\mathbf{0.0183}_{\pm0.0010}$ |

ding distribution with latent diffusion and decode samples back to dense connectomes, producing synthetic graphs that match held-out data in weight distributions, dominant eigenmodes, and graph statistics. Finally, the same latent geometry offers an interpretable coordinate system for quantifying task-evoked reconfiguration and relating both global dispersion and directional variation to cognitive performance. Overall, our framework unifies compact encoding, generation, and interpretation for dense functional connectomes.

Our framework operates on region-level connectomes with fixed node correspondence under a shared parcellation, which is the standard setting in functional connectomics; however, it limits direct application to graphs with varying node sets, partial node correspondence, or atlas mismatch. The quadratic cost of dense attention similarly restricts extension to voxel- or vertex-level resolutions. The model also relies on the induction from diffusion-map functional gradients, whose estimation depends on fMRI preprocessing and alignment to a shared template. Broader validation across additional sites, acquisition protocols, and atlases remains an important direction for future work.

## Acknowledgments

This work was supported by the Cambridge Trust and the Centre for Human-Inspired Artificial Intelligence.

## Impact Statement

This research presents an unsupervised, geometry-guided autoencoder for brain functional connectivity graphs and a latent diffusion model for sampling synthetic connectomes. Positive impacts include more sample-efficient connectome representation learning, improved tools for studying cognition and brain disorders, and enabling method development when labeled data are limited. Risks include potential cohort and preprocessing biases that could mislead group comparisons, over-interpretation for clinical utility without external validation, and privacy concerns if embeddings or generated samples retain identifiable signals. We mitigate by using de-identified datasets, evaluating on held-out subjects, and positioning our outputs as research tools. We recommend bias, robustness, and privacy audits before downstream applications.

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

## A. fMRI Data Details

**HCP** Resting-state fMRI data were available for 1067 unique subjects. Task fMRI data included working memory (WM; N=1065), emotion (N=1019), motor (N=1061), language (N=1024), social (N=1026), relational (N=1016), and gambling (N=1065); task subject indices were subsets of the resting-state cohort. We created subject-wise training/validation/test splits (70%/10%/20%) on resting-state indices and fixed them for all training and analyses (transformer autoencoder, diffusion model, and latent-space classifications), resulting in 746/107/214 subjects with no overlap. For the WM task, each subject's time series was partitioned into eight segments (2 cognitive loads × 4 stimulus types). Thus, the resting-state dataset contained one functional connectivity (FC) matrix per subject; the multi-task dataset included up to seven FC matrices per subject (one per task, when available); and the working-memory dataset contains eight FC matrices per subject, all under the same subject-wise split.

**BSNIP** Resting-state fMRI data were available for 984 individuals, including healthy controls (N=187), schizophrenia (N=172), schizoaffective disorder (N=117), psychotic bipolar disorder (N=108), and relatives of each disease group (N=163, 126, and 111). We used 5-fold cross-validation and, within each outer-fold training set, held out 10% for validation, producing five distinct 70%/10%/20% train/val/test partitions. All splits were stratified to preserve class proportions across the seven groups.

**Training-derived parcellations** We adopted data-driven parcellations with clustering (Craddock et al., 2012; Blumensath et al., 2013; Thirion et al., 2014). All fMRI time series were registered to the Glasser 360-region atlas (Glasser et al., 2016). To obtain custom parcellations (64, 100, and 150 regions) without subject leakage, we derived coarse parcellations using only *HCP resting-state training subjects*. For each training subject, we computed the $360 \times 360$ FC matrix (Pearson correlation), then formed a group-mean FC matrix by averaging Fisher $z$ values and applying the inverse transform. Using its row vectors as connectivity profiles, we applied agglomerative hierarchical clustering to merge parcels into $K \in \{64, 100, 150\}$ clusters, producing a deterministic $360 \to K$ mapping. This mapping was *fixed* and applied unchanged to all remaining HCP data (validation/test resting-state and all tasks) and to BSNIP (all folds). For each subject and scan, we averaged Glasser time series within clusters as regional time series and computed FC at the target resolution ($K \times K$).

## B. Functional Gradient Alignment and Spread

Orthogonal Procrustes alignment mapped each subject's diffusion-map gradients to a template estimated from the resting-state *training* cohort, enforcing consistent gradient orientations across subjects (Fig. 5). Fig. 6 illustrates the association between eigenvalues ($\lambda$) and gradient spread in the unaligned space, and a stability check in the aligned space comparing the raw gradient range with the quantile range $q_{0.95} - q_{0.05}$.

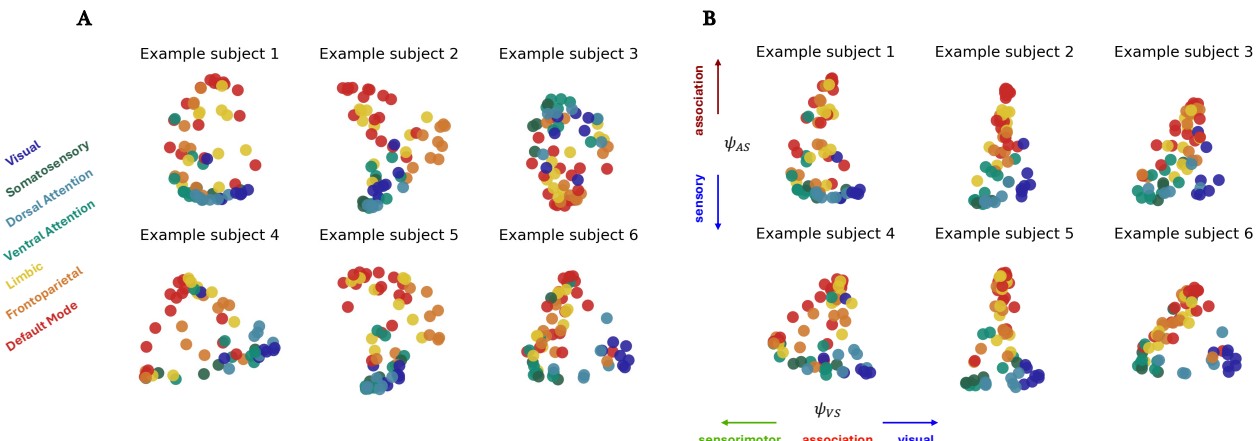

*Figure 5.* **Effect of gradient alignment.** Example subjects shown in the first two diffusion components, before (**A**) and after (**B**) orthogonal Procrustes alignment to the training-derived template. Alignment ensures cross-subject comparability, where the first two gradients represent the association-sensory ($\psi_{AS}$) and visual–sensorimotor ($\psi_{VS}$) gradients.

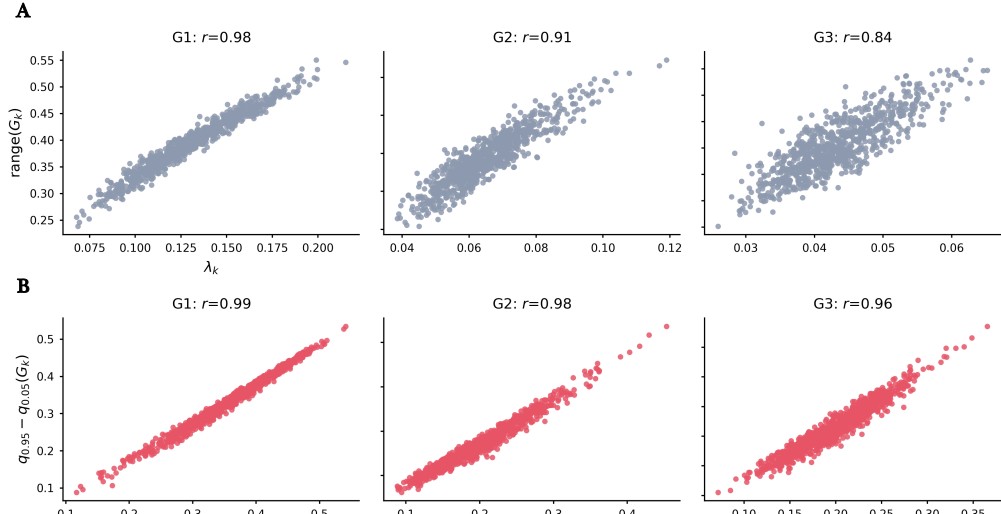

*Figure 6.* **Spread–spectrum relationships and stability.** (**A**) In the unaligned space, the gradient range shows a strong positive relationship with the diffusion spectrum $\lambda$. (**B**) In the aligned space, gradient corresponds to the quantile spread measure $q_{0.95}-q_{0.05}$, suggesting that the variability is not sensitive to outliers.

## C. Hyperparameters and Training Protocol

**Feature Normalizations** For each training pipeline (rest, WM, and task), all node features were normalized using the statistics computed on the training set only and then applied unchanged to validation and test sets. For diffusion map embeddings (aligned functional gradients), all dimensions were scaled by a single global training standard deviation to preserve the eigenvalue induced relative scaling and hierarchy across dimensions.

**Number of aligned gradients ($F$)** To assess sensitivity to the number of aligned gradients used as node features, we trained the same resting-state model under $F_{\text{rest}} \in \{5, 10, 20, 30\}$ with all other hyperparameters fixed. The resulting validation MSEs were $[0.0134, 0.0122, 0.0123, 0.0120]$, showing a broad plateau around $F_{\text{rest}} = 10$–$20$. We therefore used $F_{\text{rest}} = 10$, which captures dominant low-frequency functional geometry while avoiding unnecessary gradient dimensionality. For task-state experiments on more heterogeneous FC profiles, the main results (Section 3.2.3) show increasing $F$ from 10 to 30 improved WM reconstruction, motivating $F_{\text{task}} = 30$.

**Spatial transformer autoencoder** Implementation details follow Section 2.2 and 2.3. The encoder compresses node features $X \in \mathbb{R}^{N \times F}$ and dense, signed edge weights $C \in \mathbb{R}^{N \times N}$ to a graph-level embedding. Node features are linearly projected to hidden size $d_h$ and scalar edge weights $C_{ij}$ are linearly projected to edge embeddings in $\mathbb{R}^{d_e}$, with signed weights preserved and no thresholding applied within the model. Each encoder block uses $H$-head attention ($d_k = d_h/H$) where edge tokens modulate attention logits and values. Each block includes a 2-layer feedforward network (FFN) with GELU and dropout. For node tokens the FFN maps $d_h \rightarrow d_{ff} \rightarrow d_h$, and for edge tokens it maps $d_e \rightarrow d_{ff}^{(e)} \rightarrow d_e$. Graph-level readout uses attention pooling with a learned context vector and a linear projection to a $d_g$-dimensional graph embedding, where we used $d_g{=}16$ for resting-state and $d_g{=}32$ for task-state experiments.

The decoder reconstructs the dense $C$ using a learnable node memory table $M \in \mathbb{R}^{N \times d_m}$ (one vector per node) that defines shared keys and values across batch. Node states are first initialized from $M$ then updated by a stack of $L_d$ $H$-head cross-attention layers with hidden size $d_h$. Here, queries depend on the $z_g$ and current node state, and keys and values depend only on $M$. Each decoder layer includes a FFN mapping $d_h \rightarrow d_{ff} \rightarrow d_h$. Final node embeddings $r_i \in \mathbb{R}^{d_r}$ are produced by a multi-layer perceptron (MLP), and edges are reconstructed with an MLP applied to $[r_i, r_j, r_i \odot r_j, |r_i - r_j|, z_g]$. Symmetry is enforced by $\frac{1}{2}(\tilde{C} + \tilde{C}^\top) \rightarrow \hat{C}$.

We used $L_e{=}4$, $d_h{=}48$, $d_e{=}2$, $H{=}4$, $d_{ff}{=}64$, $d_{ff}^{(e)}{=}16$, and $p{=}0.2$ for the encoder. For the decoder we used $L_d{=}2$, $H{=}4$, $p{=}0.2$, $d_{ff}{=}128$, and memory cross-attention dimensions $(d_m, d_h, d_r)$ of $(32, 32, 32)$ (resting-state) or $(64, 32, 64)$ (task-state).

**Optimization and model selection**   We optimized dense FC reconstruction with mean-squared error (MSE) over all entries (including the diagonal), $\mathcal{L} = \text{MSE}(C, \hat{C})$. Models were trained with Adam (betas $(0.9, 0.95)$) for up to 150 epochs, and we selected the checkpoint with the lowest validation loss using early stopping with patience of 30 epochs. We used `ReduceLROnPlateau` for learning rate on validation loss, with factor = 0.5, patience = 10 epochs, threshold = $10^{-4}$, minimum learning rate = $10^{-5}$. Resting-state training used learning rate $2 \times 10^{-3}$ and batch size 8; task-state training used learning rate $3 \times 10^{-3}$ and batch size 64.

**Neural dynamics extension (working memory)**   For WM time series $Y_{1:T} \in \mathbb{R}^{N \times T}$, we model $Y_{1:T}$ as a sequence of whole-brain activation vectors and encode it with a GRU whose hidden size was $H_g{=}128$. Hidden states were aggregated by temporal mean to form the dynamics summary, which was fused with the spatial encoder embedding to obtain a joint spatial-temporal embedding with dimension $d_g{=}32$. For temporal decoding, we used initial-condition and context conditioning described in Section 2.5, where the decoder GRU unrolled an $r$-dimensional latent trajectory with $r{=}8$, followed by a linear readout to reconstruct $\hat{Y} \in \mathbb{R}^{N \times T}$. For the temporally extended pipeline, the training objective consisted of two components, time-series reconstruction MSE between $(Y, \hat{Y})$ and FC reconstruction MSE between $(C, \hat{C})$. Both terms were combined with equal fixed weights. The time series term is computed on per-region normalized signals (zero mean and unit variance), while the FC term is computed on the dense FC matrix in its native scale.

**Latent diffusion**   After training the autoencoder, we fit a denoising diffusion probabilistic model (Ho et al., 2020) on the normalized graph embeddings $z_g \in \mathbb{R}^{d_g}$, where normalization used the training-set mean and standard deviation $(\mu_{\text{train}}, \sigma_{\text{train}})$. We adopted a linear noise schedule with $T{=}1000$ steps and $\beta_t \in [10^{-4}, 2{\times}10^{-2}]$. The denoiser $\epsilon_\theta(\tilde{z}_t, t)$ was constructed as an MLP conditioned on a 128-dimensional time embedding, with hidden dimensions $(128, 256, 256, 128)$ and LeakyReLU activations. The model was trained for 1000 epochs with Adam (learning rate $10^{-3}$) by minimizing MSE on the diffusion target (the added noise $\epsilon$) and selecting the checkpoint with the lowest validation loss. During sampling, the autoencoder decoder was kept frozen. The sampled latents were de-normalized using $(\mu_{\text{train}}, \sigma_{\text{train}})$ and decoded to dense FC graphs.

**Computational cost**   We profiled per-training-step runtime on HCP resting-state data with batch size 8 on a single GPU. Our model required 26.4 ms per step, compared with 6.9 ms for GAE, 23.4 ms for Graphite, 32.9 ms for GATE, and 71.8 ms for GRALE. The cost is therefore moderate across reconstruction baselines in this dense connectome regime, while resulting in the lowest reconstruction error (Table 1).

# D. Baseline Models

All baselines were adapted to produce deterministic graph-level embeddings $z_g$ and to reconstruct the dense FC matrices where applicable. Aligned functional gradients were used as node attributes for all models.

**Graphite**   Graphite (Grover et al., 2019) is a latent variable framework in which decoding entails a reverse message passing process. From node latent representations $Z$, the decoder first constructs an intermediate graph from an inner product and then refines $Z$ with message passing, iterating this procedure for multi-step refinement. In the Graphite-AE variant, the model minimizes adjacency reconstruction error. In our adaptation, which requires a compact graph embedding, we used a graph convolutional network (GCN) encoder with attention pooling to obtain $z_g$, linearly projected $z_g$ to an initial node-level representation $Z^{(0)}$, from which the Graphite refinement process was ran to produce $Z^*$ and reconstruct $\hat{C}$ by inner product.

**GRALE**   GRALE (Krzakala et al., 2025) is a graph-level autoencoder designed for variable-sized graphs. It jointly encodes node and pair (edge) representations with an Evoformer module (Jumper et al., 2021), and pools the learned pairwise representations into a small set of latent graph tokens. Decoding uses Evoformer decoder conditioned on the latent tokens. The original method further introduces a differentiable node-matching module and an Optimal Transport (OT)-inspired reconstruction objective for addressing node correspondence. In our connectome setting, node correspondence and graph size are fixed by parcellation. Hence, we retained the Evoformer encoder, pooling, and decoder, but removed the matching and OT components and trained with direct MSE loss on the dense FC matrices.

**GATE**   GATE (Liu et al., 2021) is a connectome-specific variational graph autoencoder. The inference network encodes each connectome (vectorized edges) into a Gaussian latent using an MLP, and the generative model decodes $z_g$ into node-wise factor vectors, and reconstructs connectivity by aggregating the corresponding factor-wise outer products. The original

design for structural connectomes uses a Poisson likelihood for count valued connectivity. In our dense FC adaptation, edges are continuous and signed, hence we used MSE on $\hat{C}$ and the deterministic path $z_g = \mu$ with no resampling.

**Representation-only**    To assess latent separability without graph decoding, we included two representation-only controls. FC+UMAP vectorizes each FC matrix and applies UMAP to obtain a low-dimensional embedding for downstream logistic regressions; `n_neighbors` $\in \{15, 30, 60\}$ and `min_dist` $\in \{0.0, 0.1, 0.3, 0.5\}$ were tuned on the validation set (Euclidean metric). GraphMAE (Hou et al., 2022) is a self-supervised masked graph autoencoder trained to reconstruct masked node features. In our adaptation, we used a weighted GCN encoder–decoder and obtained $z_g$ by pooling the encoder node embeddings for downstream classifiers.

## E. Code and Data Availability

Code is available at github.com/SubatA20/geometry-guided-brain-graph-AE. The Human Connectome Project Young Adult (HCP-YA) data are available through the Human Connectome Project's data portal (ConnectomeDB) subject to the HCP data use terms and required registration. The Bipolar and Schizophrenia Network for Intermediate Phenotypes (BSNIP) data are available via the NIMH Data Archive (NDA) under controlled access.

