# OpenReview forum: "Geometry-Guided Generative Representation for Functional Brain Graphs"
_ICML.cc/2026/Conference — ICML 2026 regular_

### Official Review · Reviewer_F1JS · 2026-02-28

**Soundness:** 3
**Presentation:** 3
**Significance:** 3
**Originality:** 3
**Overall Recommendation:** 4
**Confidence:** 3

**Summary:**

This paper proposes an unsupervised deep learning framework for learning compact graph-level representations of functional brain connectomes. The manuscript intends to consider the central issue of modeling dense, weighted functional connectivity (FC) graphs without the information loss associated with standard sparsification or thresholding techniques. The authors introduce a geometry-guided graph transformer autoencoder that utilizes aligned functional gradients as node features to provide a strong inductive bias. The resulting latent space is shown to capture topological properties, differentiate cognitive states, and predict clinical status (Schizophrenia vs. Healthy Controls). Furthermore, the authors extend the framework with a latent diffusion model to generate synthetic connectomes and a recurrent neural network to model temporal dynamics. Overall, the paper's important contribution concerns the unification of spectral graph theory (functional gradients) with modern generative deep learning to produce interpretable and robust embeddings for neuroimaging data.

**Compliance With Llm Reviewing Policy:**

Affirmed.

**Final Justification:**

The authors addressed my concerns, I maintain my score.

**Key Questions For Authors:**

1. Why was Latent Diffusion chosen over simpler generative models like VAEs or GANs on the latent space? Is the complexity of diffusion necessary for the low-dimensional $z_g$ (dim=16 or 32)?
2. In the BSNIP dataset (Schizophrenia/Bipolar), functional gradients might deviate significantly from the healthy template. Does enforcing Procrustes alignment to a healthy resting-state template mask potential pathological signal that manifests as geometric distortion?

**Limitations:**

1. The model operates on a fixed parcellation. It cannot easily adapt to datasets with different numbers of regions or different atlases without retraining and redefining the alignment template.
2. The quadratic complexity of the transformer limits the resolution of the brain graphs that can be processed.

**Strengths And Weaknesses:**

### Strengths:
1. The integration of functional gradients (eigenvectors of the graph Laplacian/normalized adjacency) as node features is highly motivated by neuroscience literature (Margulies et al.). This effectively bridges classical spectral analysis with modern deep learning, allowing the model to respect the intrinsic geometry of the brain.
2. Most Graph Neural Networks (GNNs) struggle with dense graphs (over-smoothing). By using an edge-conditioned transformer and avoiding message-passing aggregation in favor of global attention, the authors successfully model the full weighted connectome. The theoretical motivation and architectural choices (e.g., cross-attention decoder with node memory) are well-justified for this specific data modality.
3. The paper evaluates the method across multiple dimensions: reconstruction quality, latent space separability (rest vs. task), downstream clinical classification (BSNIP dataset), and generative quality. The comparison against a wide range of baselines (GAE, Graphite, GRALE, GATE) strengthens the claims.
4. Successfully training a diffusion model on the latent space $z_g$ rather than the high-dimensional matrix space is an efficient approach. The validation showing that generated graphs preserve spectral and topological properties (Figure 4) is strong.

### Weakness
1. The method assumes a fixed set of nodes (parcellation) and relies on Procrustes alignment of gradients to a template. While this works for the datasets used, it may limit the model's ability to handle severe pathologies where the functional geometry is radically altered or situations where node correspondence is not perfect across subjects.
2. While the application and combination are novel, the individual components (Graph Transformers, Latent Diffusion, RNNs for dynamics) are standard. The novelty lies primarily in the specific orchestration for fMRI data rather than a fundamental machine learning algorithmic advance.
3. The method uses full attention $N \times N$. While fine for $N=360$, this approach does not scale to voxel-level or high-resolution vertex-level analysis, which is a direction of interest in the field.

---

> ### Author Rebuttal · Authors · 2026-03-30
>
> Thank you for the thoughtful review, and for recognizing the value of integrating functional gradients, edge-conditioned and node-memory attention, and latent-space generation for functional connectomes. We are encouraged that the framework is viewed as technically solid and has potential for the field. Our responses are below.
>
> **On latent diffusion versus latent VAE or GAN**
>
> Thank you for this important point. This choice is driven more by the flexibility of the latent distribution model than by the dimensionality of $z_g$. Even in lower dimensions, the empirical latent distribution can be heterogeneous and multi-modal across subjects, cognitive states, and clinical groups. Standard VAEs optimize a variational objective under an explicit prior, typically Gaussian, whereas diffusion provides a more flexible way to model latent distributions without imposing that assumption. It also avoids adversarial optimization and the associated instability and mode collapse issues of GANs [1]. The computational cost is also modest in our setting given that diffusion is applied to compact graph-level embeddings rather than the full $N^2$ connectome, with a small MLP denoiser operating on $z_g \in \mathbb{R}^{16}$ or $\mathbb{R}^{32}$. This is in line with the motivation for latent diffusion broadly [2] and for recent latent graph diffusion models [3]. Hence, while we do not claim diffusion is the only viable option, we chose it because it offers a strong balance of flexibility and stability for dense connectome generation.
>
> **On gradient alignment and pathological signal**
>
> We appreciate this important concern. In our framework, Procrustes alignment is applied only to resolve the sign, order, and rotation ambiguity in subject-level gradients. For instance, a subject’s first gradient may correspond to the group’s second, and its direction may be flipped. Alignment is therefore necessary for meaningful cross-subject comparison. With this transformation, the intrinsic within-subject geometry is not altered in the retained gradient subspace, and the procedure serves to standardise the coordinate system. Pathology-related effects manifested as altered spread, gradient contraction or expansion, or displacement within the aligned space can therefore still be represented by the model. In gradient literature, recent work comparing template strategies shows that alignment to an independent HCP template improves robustness of between-group effects rather than suppressing it in clinical conditions [4]. A recent gradient study on schizophrenia likewise leverages an HCP-derived reference to avoid dataset-specific bias [5]. In our experiments, the reference template is derived strictly from the resting-state training set to avoid leakage, alignment improves performance in ablations, and aligned gradients support BSNIP latent classification. That said, we agree on the limitation that if pathology induces a substantial basis reorganization that is poorly captured by the shared template, or if node correspondence is imperfect, the fixed-template approach may attenuate some signals. We will make this caveat more explicit in the discussion.
>
> **On novelty, fixed parcellation, and scalability**
>
> We appreciate this framing and agree that our contribution is the connectome-specific integration of established building blocks, with graph-level learning for dense weighted FC without sparsification, guided by aligned functional gradients, and a unified latent space supporting reconstruction, fusion with dynamics, generation, and interpretation. We also agree that the current framework is intended for region-level connectomes with fixed node correspondence under a shared parcellation. This remains the standard setting in most functional connectomics and fMRI graph-learning pipelines, although it limits transferability across atlases and to partial node graphs. Similarly, quadratic attention limits direct extension to voxel- or vertex-level resolution. These are important directions for extending this line of work, and we will make our intended scope and these limitations clearer in the revision.
>
> **References**
>
> [1] Metz et al. Unrolled Generative Adversarial Networks. *ICLR*, 2017.
>
> [2] Rombach et al. High-Resolution Image Synthesis with Latent Diffusion Models. *CVPR*, 2022.
>
> [3] Zhou et al. Unifying Generation and Prediction on Graphs with Latent Graph Diffusion. *NeurIPS*, 2024.
>
> [4] Kim et al. Comparison of Different Group-Level Templates in Gradient-Based Multimodal Connectivity Analysis. *Network Neuroscience*, 2024.
>
> [5] Shevchenko et al. A Comparative Machine Learning Study of Schizophrenia Biomarkers Derived from Functional Connectivity. *Scientific Reports*, 2025.

---

> > ### Author Rebuttal · Reviewer_F1JS · 2026-04-01
> >
> > Thanks for the clarification, I'll keep my score.

---

### Official Review · Reviewer_mwUf · 2026-03-05

**Soundness:** 2
**Presentation:** 3
**Significance:** 2
**Originality:** 2
**Overall Recommendation:** 4
**Confidence:** 4

**Summary:**

The paper presents an unsupervised graph transformer autoencoder for learning representations of functional brain connectivity graphs. The model encodes connectomes into a compact latent vector and reconstructs the connectivity matrix. The learned embeddings are evaluated through reconstruction, downstream classification tasks, and a latent diffusion model used to generate synthetic brain graphs.

**Compliance With Llm Reviewing Policy:**

Affirmed.

**Final Justification:**

The rebuttal addresses my main concerns, and I would like to increase my rating to weak accept.

**Key Questions For Authors:**

See above.

**Limitations:**

Partially. The paper includes limited discussion of limitations. It would be helpful to discuss the higher model complexity compared to baselines, the reliance on predefined node features, and the indirect generative process based on latent diffusion. A short discussion of these limitations and possible future work in the discussion or conclusion section would strengthen the paper.

**Strengths And Weaknesses:**

The method is technically reasonable and the reconstruction objective is appropriate for unsupervised learning. Experiments on neuroimaging datasets and comparisons with several baselines generally support the claims. The paper is mostly clear and the overall pipeline of the encoder, decoder, and latent diffusion model is generally easy to follow.

However, the paper does not analyze computational complexity or scalability compared with the baselines, and the proposed model seems more complex than several of the compared methods. The training procedure is unsupervised, but part of the evaluation relies on downstream classifiers trained on the learned embeddings, and the details of this evaluation pipeline are not fully described. Some components used in the experiments, such as the classifiers for downstream tasks and parts of the generative pipeline, are not clearly integrated into the provided code, which may affect reproducibility.

In addition, the method mainly combines existing techniques such as graph transformers, autoencoders, and diffusion models, so the methodological novelty appears moderate. The generative component is based on diffusion in the latent space rather than directly modeling graph generation, which makes the generative aspect somewhat indirect.

Although the paper discusses the latent space in relation to brain network properties, it lacks deeper interpretability analysis of the model itself. For example, the paper does not analyze attention patterns in the transformer, identify which brain regions or connections are most influential, or examine how perturbations of the connectivity matrix affect the learned representations. Such analyses could help better understand how the model captures the structure of brain networks.

There are also some minor inconsistencies in notation (for example the reuse of symbols such as "W" for both connectivity matrices and learnable parameters), which may cause confusion when following the equations.

---

> ### Author Rebuttal · Authors · 2026-03-30
>
> We sincerely thank the reviewer for the careful assessment and feedback. We appreciate the positive comments on the technical soundness and experimental evaluation. Here are our point-by-point responses.
>
> **On computational complexity and scalability**
>
> We agree the manuscript would benefit from clearer complexity comparison and will include this in the revision. Our design is motivated by the dense, weighted connectome setting, where sparsification may remove meaningful connections and repeated message passing can risk over-smoothing; we therefore use dense edge-conditioned attention. Table 2 confirms the full model outperforms ablations across resolutions. We profiled reconstruction-enabled baselines on HCP resting-state data (batch size 8, single GPU). Our model requires 26.4 ms per training step, versus 6.9 ms for GAE, 23.4 ms for Graphite, 71.8 ms for GRALE, and 32.9 ms for GATE. The computational cost is therefore moderate in the dense connectome regime.
>
> **On unsupervised training and downstream classifiers**
>
> The representation learning is fully unsupervised, the autoencoder is trained only to minimize reconstruction loss, without using task or diagnostic labels. Downstream classifiers are used only as post hoc evaluations of the learned graph-level embedding $z_g$, to test whether the compressed representation preserves functionally meaningful structure. Specifically, we freeze the learned embeddings after training and fit a simple logistic regression classifier on the training set, then evaluate on the test set. No classifier-based supervision is used to train the representation, and no hyperparameter tuning is performed for the linear models. The two evaluations are complementary; reconstruction error measures how well the unsupervised objective captures dense FC structure, while logistic regression on frozen embeddings tests whether the same low-dimensional representation retains functionally or clinically relevant variation. We will clarify this pipeline in the revision.
>
> **On methodological contribution**
>
> We agree that graph transformers, autoencoders, and diffusion models are established components. Our contribution is a connectome-specific formulation for dense, weighted functional connectomes, where standard graph-learning pipelines are limited. Existing approaches often rely on sparsification that can alter FC geometry, or on node-level embeddings rather than compact graph-level representations. We address this with a graph-level autoencoding framework guided by aligned functional gradient geometry as an inductive bias. The resulting latent space unifies dense connectome reconstruction, integration with neural dynamics, generation, and interpretable analysis of task-induced brain reconfiguration. Our ablations show that alignment, edge conditioning, memory-based decoding, and gradient geometry each contribute to performance across settings.
>
> **On the generative component**
>
> We use latent diffusion because direct graph-space generation for dense, weighted functional connectomes would require modelling $N^2$ edge variables and their complex dependencies. Therefore, we learn a compact graph-level latent representation, fit diffusion in this low-dimensional space, and decode samples back to full FC graphs. This is in line with the motivation for latent diffusion broadly [1] and for recent latent graph diffusion models [2]. This keeps generation aligned with the same latent geometry used for representation learning and interpretation. Generation is evaluated across latent (MMD, 1-NN), matrix (weight distribution), and graph spaces (eigenvalues, clustering, modularity, small-worldness).
>
> **On interpretability analysis**
>
> We agree that region-level attention or perturbation analyses could provide a local view of the model. Our interpretability focus here is intentionally global and graph-level, to assess whether the learned low-dimensional embedding preserves large-scale connectome organization across subjects and conditions. Accordingly, we examine whether the latent space captures structured variation in functional-gradient geometry and graph-theoretic organization, and whether variation along latent directions relates to cognition. We will clarify this emphasis in the revision.
>
> **On code integration, notation and limitations**
>
> We appreciate these comments. The supplementary code package covers the main experiments and appendix details. In the revision, we will add the latent-space classifiers and generative pipeline, clean up notation (including the reuse of $W$), and expand the limitations discussion, including fixed parcellation and reliance on predefined functional gradients as node features, which may influence downstream results.
>
> **References**
>
> [1] Rombach et al. High-Resolution Image Synthesis with Latent Diffusion Models. *CVPR*, 2022.
>
> [2] Zhou et al. Unifying Generation and Prediction on Graphs with Latent Graph Diffusion. *NeurIPS*, 2024.

---

> > ### Author Rebuttal · Reviewer_mwUf · 2026-04-02
> >
> > The rebuttal addressed my main concerns; I update my score to 4 (Weak Accept).

---

### Official Review · Reviewer_ypaL · 2026-03-08

**Soundness:** 3
**Presentation:** 4
**Significance:** 4
**Originality:** 3
**Overall Recommendation:** 5
**Confidence:** 3

**Summary:**

The paper proposes a geometry-guided graph transformer autoencoder to learn compact latent representations of dense functional brain connectivity graphs. The model incorporates aligned functional gradients as node features, providing a biologically meaningful inductive bias. The encoder produces a graph-level latent embedding, which is decoded through a memory-based cross-attention mechanism to reconstruct the full connectivity matrix. The learned latent space preserves interpretable properties of brain organization and enables unsupervised decoding of cognitive states, which improves further when neural dynamics are incorporated through a temporal extension. To generate synthetic brain graphs, the authors train a diffusion model in the latent space and decode sampled embeddings into connectivity matrix. Experiments on HCP and BSNIP datasets show improved reconstruction accuracy, meaningful latent geometry related to cognition, and realistic generation of functional connectivity graphs.

**Compliance With Llm Reviewing Policy:**

Affirmed.

**Final Justification:**

Overall, I enjoyed reading this paper. I believe it offers meaningful contributions, and for this reason, I recommend that it be accepted.

**Key Questions For Authors:**

### 1. Clarification of Figure 4B
It is not entirely clear what Figure 4B represents. Could the authors clarify what the scatter plot shows and how it should be interpreted?

### 2. Fixed graph size assumption
The model assumes a fixed number of nodes. This design choice may limit the applicability of the method to datasets where connectivity graphs contain partial observations or different numbers of nodes. Do the authors foresee a way to extend the framework to reconstruct partial connectomes or graphs with a variable number of nodes?

### 3. Generalization across datasets
The experiments focus primarily on the datasets used in the paper. Would it be possible to evaluate the model on connectivity graphs from additional datasets to assess the generalizability of the learned geometry-guided latent space across different settings?

**Limitations:**

The paper does not explicitly discuss the limitations of the proposed method. In particular, the model assumes a fixed number of nodes corresponding to a predefined brain parcellation. This assumption may limit its applicability to settings where connectivity graphs have partial or missing regions, or where the number of nodes varies across datasets.

**Strengths And Weaknesses:**

## Soundness:
The submission appears technically sound and the experimental results support the claims made in the paper. The methodology is clearly described and the empirical evaluation is reasonably comprehensive. One possible improvement would be to evaluate the learned representation on a truly different dataset or acquisition setting, which would help demonstrate that the geometry-guided latent space generalizes beyond the specific datasets considered and is not overly tied to a single experimental setup.

## Presentation:
The paper is well written and generally easy to follow. The introduction provides clear motivation and the work is well positioned with respect to prior literature. The methods section is detailed and helps the reader understand the architectural choices and training procedure, with reproducible details. One aspect that could be improved concerns the figures, as some of them would benefit from additional explanation. For example, in Figure 4B it is not entirely clear what the scatter plot represents or how it should be interpreted in relation to the generation results.


## Significance:
The paper addresses an important problem in brain connectivity analysis and representation learning. The proposed framework provides an interesting perspective on how geometric priors can guide representation learning. If validated more broadly, this approach could influence future work on connectome representation learning and generative modeling of brain graphs.


## Originality:
The paper introduces a novel approach for learning representations of brain connectivity graphs by incorporating geometry-guided inductive biases through aligned functional gradients. This is a creative and well-motivated idea that provides new insights into how large-scale brain organization can be leveraged in graph representation learning. The integration of a latent diffusion model for generating synthetic connectomes is also an interesting addition, although the paper could benefit from providing more qualitative examples or analysis of the generated graphs.

---

> ### Author Rebuttal · Authors · 2026-03-30
>
> We sincerely appreciate the reviewer’s endorsement and thoughtful assessment, and for highlighting the paper’s potential significance for connectome representation learning.
>
> **On Figure 4B**
>
> We agree this panel should be clarified. The bottom row in panel B shows two representative dense FC matrices decoded from sampled latent points in regions of lower and higher association-sensory gradient ($\psi_{AS}$) range, corresponding to the cyan and magenta stars in panel A. The top row shows the diffusion-map embeddings recomputed from those generated matrices. The low $\psi_{AS}$ range example (left) shows weaker association-sensory differentiation, reflected by a more compressed gradient embedding and a more homogeneous FC pattern when nodes are ordered along the association-to-sensory axis. The high $\psi_{AS}$ range example (right) shows stronger association-sensory separation, with a more expanded gradient embedding and a correspondingly more modular connectivity pattern. The panel is therefore intended to illustrate that movement in latent space corresponds to systematic variation in both the generated connectome and recovered functional geometry. We will rephrase and extend the caption to clarify the interpretation. Figure 4 and Section 3.4 provide complementary generative evaluations across latent space, weight distributions, and spectral and topological properties.
>
> **On fixed graph size and partial connectomes**
>
> We agree the current framework is designed for fixed node correspondence under a parcellation, which is the standard region-level setting in many fMRI connectomics pipelines and the intended scope of this paper. Extending the framework beyond this setting is an important future direction. For variable-sized graphs, one possible route is graph autoencoding with size handling and permutation-invariant reconstruction [1]. For partial connectomes, partial graph matching or optimal partial transport, which allow unmatched nodes rather than forcing full correspondence, are promising directions [2]. Moreover, recent brain-graph pretraining across multiple atlases with varying parcellations suggests a route toward more atlas-robust representations [3]. We will clarify this scope and these future extensions in the revision.
>
> **On generalization across datasets and settings**
>
> We agree that broader external validation would strengthen the paper. The current evaluation spans HCP resting-state, HCP task-state, and BSNIP, providing evidence across resting, task, and clinical regimes. However, this is not a dedicated cross-acquisition generalization experiment. We will make this clearer and note that broader validation across additional datasets, sites, and acquisition settings remains an important next step.
>
> **On limitations**
>
> We appreciate this suggestion and will include a clear limitation section in the revision. This will include the current design for region-level connectomes with fixed node correspondence, which limits direct applicability to partial connectomes or graphs with varying node sets. We will also note that broader validation across additional datasets and acquisition settings remains an important direction for future work.
>
> **References**
>
> [1] Winter et al. Permutation-Invariant Variational Autoencoder for Graph-Level Representation Learning. NeurIPS, 2021.
>
> [2] Ratnayaka et al. Learning Partial Graph Matching via Optimal Partial Transport. ICLR, 2025.
>
> [3] Wei et al. A Brain Graph Foundation Model: Pre-Training and Prompt-Tuning across Broad Atlases and Disorders. ICLR, 2026.

---

> > ### Author Rebuttal · Reviewer_ypaL · 2026-04-01
> >
> > I thank the authors for the clarifications. I confirm my score for acceptance

---

### Official Review · Reviewer_bYrb · 2026-03-12

**Soundness:** 3
**Presentation:** 3
**Significance:** 3
**Originality:** 3
**Overall Recommendation:** 4
**Confidence:** 4

**Summary:**

This paper introduces aligned functional gradients as an inductive bias for functional brain graph representation learning, providing a novel framework that addresses the limitations of previous graph learning methods in brain connectome analysis. The authors make the following key contributions:

1. They propose a customized Transformer architecture for dense, weighted FBGs, which bridges the gap between generic graph Transformer self-attention mechanisms and the underlying structural information of graphs.

2. They design a unified framework for both representation learning and generative modeling, enabling a more comprehensive and coherent treatment of FBG data within a single paradigm.

**Compliance With Llm Reviewing Policy:**

Affirmed.

**Final Justification:**

The rebuttal has addressed my concerns, and thus I maintain my score.

**Key Questions For Authors:**

1. In Section 3.1, the authors state: “We use the first $F_{\text{rest}} = 10$ aligned functional gradients ...”. What would happen if $F_{\text{rest}}$ were set to other values? An analysis of the sensitivity of the model to this hyperparameter would strengthen the paper.

2. Section 2.5 introduces the spatiotemporal extension, which includes both the mean squared error (MSE) for time-series reconstruction and the MSE for functional connectivity (FC) reconstruction. However, the paper does not clearly specify whether these two loss terms are equally weighted, whether they are normalized, or whether their coefficients were tuned during training.

**Limitations:**

The paper should discuss the limitations of the proposed method, such as its sensitivity to preprocessing steps and its scalability or adaptability to other generative models.

**Strengths And Weaknesses:**

**Strengths:**
This paper presents an effective unsupervised representation learning method for dense weighted functional brain graphs. The manuscript is generally well written, the methodological design is sound, and the experiments are sufficiently detailed to demonstrate the effectiveness of the proposed approach.

**Weaknesses:**

1. Although the manuscript provides detailed descriptions of the encoder, decoder, and latent diffusion modules, it does not include an overall visualization of the proposed framework. The authors are encouraged to provide either an architectural diagram or algorithmic pseudocode to help readers better understand the complete pipeline.

2. The paper does not provide a systematic review of prior studies on geometry-guided graph representation learning, nor does it clearly clarify how the proposed method differs from existing geometry-guided approaches.

---

> ### Author Rebuttal · Authors · 2026-03-30
>
> We sincerely thank the reviewer for their thoughtful, positive evaluations and constructive suggestions. Here are our point-by-point responses.
>
> **On previous geometry-guided studies**
>
> We thank the reviewer for this point. We agree that the distinction from prior geometry-guided graph learning can be made more clearly. Related work can be grouped into (1) general graph transformers and GNNs that inject structure with positional or structural encodings (e.g., shortest-path and centrality biases, random-walk statistics, and learnable embeddings) [1,2]; (2) geometric graph learning using external spatial geometry (e.g., 3D coordinates, distances, angles) [3,4]; and (3) methods that define the representation space itself in non-Euclidean geometry [5].
>
> Our framework is different from these approaches in three ways. First, the geometry we use is connectome-specific and functionally derived, where we compute diffusion-map functional gradients directly from each subject's FC matrix, rather than using generic graph structure or external spatial coordinates. Second, this geometry is subject-specific yet aligned to a shared population space via Procrustes alignment to a train-only resting-state template. This produces a biologically grounded coordinate system that preserves individual variability while enabling cross-subject comparability and avoiding leakage. In our ablation (Table 2), we show that removing this alignment degrades learning. Lastly, our aim is not generic graph prediction, but a unified framework for dense functional brain graphs. The aligned functional geometry serves as an inductive bias for graph-level unsupervised representation learning, dense FC reconstruction and generation, and interpretable analysis linking spectral and graph-theoretic variations in the same latent space. We will revise the introduction to make this positioning clearer.
>
> **On the choice of the number of aligned gradients**
>
> Thank you for this suggestion. The manuscript includes a task-state (more heterogeneous FC profiles) comparison showing that increasing the number of gradients from $F=10$ to $F=30$ improved WM reconstruction. For resting-state, we conducted a small sweep with $F_{\mathrm{rest}} \in \{5,10,20,30\}$ under the same training protocol. The resulting validation MSEs were $[0.0134, 0.0122, 0.0123, 0.0120]$, showing a broad plateau around $F_{\mathrm{rest}}=10$-$20$. Therefore, we use $F_{\mathrm{rest}}=10$, which captures dominant low-frequency functional geometry while avoiding unnecessary gradient dimensionality. We will add this sensitivity analysis to the revision.
>
> **On spatiotemporal loss**
>
> Thank you for pointing out this detail. For the temporal extension, the training used equal fixed weights:
> $$
> L_{\mathrm{st}}=\mathrm{MSE}(Y,\hat{Y})+\mathrm{MSE}(W,\hat{W}).
> $$
> There was no coefficient tuning in optimization. The time-series term is computed on per-region normalized signals, while the FC term is computed on the dense FC matrix in its native correlation scale. We agree that this should be specified, and we will add this detail in Appendix Section C.
>
> **On the pipeline visualization**
>
> We agree that a pipeline visualization would be beneficial. We will add an architecture figure linking the components---gradient extraction and alignment, edge-conditioned encoder, memory-based decoder, temporal extension, and latent diffusion---in the revision to summarize the full pipeline.
>
> **On limitations**
>
> We agree that the limitations should be expanded. We will make the following points more explicit in our discussion: (1) sensitivity to preprocessing of fMRI data; (2) reliance on a fixed parcellation across subjects; and (3) the computational scalability trade-off of dense attention and brain graph decoding. Some of these are already implicit in the current writing, but we agree that a clearer discussion would strengthen the paper.
>
> We thank the reviewer again for these helpful suggestions.
>
> **References**
>
> [1] Rampášek et al. Recipe for a General, Powerful, Scalable Graph Transformer. *NeurIPS*, 2022.
>
> [2] Ma et al. Graph Inductive Biases in Transformers without Message Passing. *PMLR*, 2023.
>
> [3] Liu et al. Pre-training Molecular Graph Representation with 3D Geometry. *ICLR*, 2022.
>
> [4] Fang et al. Geometry-enhanced molecular representation learning for property prediction. *Nature Machine Intelligence*, 2022.
>
> [5] Chami et al. Hyperbolic Graph Convolutional Neural Networks. *NeurIPS*, 2019.

---

> > ### Author Rebuttal · Reviewer_bYrb · 2026-04-02
> >
> > Thank you for these responses.  I maintain my score.

---

### Decision · Program_Chairs · 2026-04-30

**Decision:**

Accept (regular)

**Comment:**

This paper proposes a geometry guided graph transformer autoencoder to learn interpretable representations of functional brain graphs. The reviewers highlighted the method's innovation, e.g. in integrating functional gradients and connecting the self-attention mechanism with the underlying graph information. The paper also provides an effective evaluation of the method.

After the rebuttal and discussion period, all reviewers lean towards accepting this paper. None of the reviewers have lingering, major concerns with the paper. Overall, the paper is technically sound, well-written, and makes an important contribution that is relevant to at least  part of the ICML community. The authors should make sure to incorporate all of the new results in the camera ready version of the paper.